# Whole genomic analysis reveals atypical non-homologous off-target large structural variants induced by CRISPR-Cas9-mediated genome editing

Hsiu-Hui Tsai[1], Hsiao-Jung Kao[2], Ming-Wei Kuo[1], Chin-Hsien Lin [3],
Chun-Min Chang[1], Yi-Yin Chen[1], Hsiao-Huei Chen[2], Pui-Yan Kwok[2,4],
Alice L. Yu [1,5,6] & John Yu [1,7] ✉

CRISPR-Cas9 genome editing has promising therapeutic potential for genetic diseases and cancers, but safety could be a concern. Here we use whole genomic analysis by 10x linked-read sequencing and optical genome mapping to interrogate the genome integrity after editing and in comparison to four parental cell lines. In addition to the previously reported large structural variants at on-target sites, we identify heretofore unexpected large chromosomal deletions (91.2 and 136 Kb) at atypical non-homologous off-target sites without sequence similarity to the sgRNA in two edited lines. The observed large structural variants induced by CRISPR-Cas9 editing in dividing cells may result in pathogenic consequences and thus limit the usefulness of the CRISPR-Cas9 editing system for disease modeling and gene therapy. In this work, our whole genomic analysis may provide a valuable strategy to ensure genome integrity after genomic editing to minimize the risk of unintended effects in research and clinical applications.

The induced pluripotent stem cell (iPSC) technology is a powerful platform for pathogenesis studies, drug screening, tissue engineering, and cell replacement therapy. Human iPSCs will enable the development of autologous, patient-specific stem cell therapies, resulting in long-term engraftment without needing immunosuppressive treatments for patients. Several transplantations of autologous iPSC-derived cells have been reported[1–4]; none of the patients in these studies has suffered severe adverse events, suggesting promising feasibility for personalized regenerative medicine. However, using iPSCs for allogeneic transplantation will require establishing large numbers of iPSC lines to cover the MHC diversities for their off-the-shelf applications in replacement cell therapy.

An alternative strategy is to develop a hypoimmunogenic single iPSC from a common donor and bank it as an off-the-shelf product for allogeneic applications[5,6]. Recently, several studies showed that human iPSCs lost immunogenicity when MHC class I and II genes were inactivated, and CD47 was overexpressed by CRISPR-Cas9 (Clustered Regularly Interspaced Short Palindromic Repeats/CRISPR-associated protein 9) genome editing[7–10].

The CRISPR-Cas9 system is a powerful technology for precise gene editing of the genome[11–14]. It consists of Cas9 endonuclease and single-stranded guide RNA (sgRNA). The sgRNA recruits the Cas9 endonuclease to cleave both DNA strands at the target site in a sequence-specific manner to generate a double-stranded break (DSB).

[1]Institute of Stem Cell and Translational Cancer Research, Chang Gung Memorial Hospital at Linkou, Taoyuan, Taiwan. [2]Institute of Biomedical Sciences, Academia Sinica, Taipei, Taiwan. [3]Department of Neurology, National Taiwan University Hospital and School of Medicine, Taipei, Taiwan. [4]Cardiovascular Research Institute, Institute for Human Genetics, and Department of Dermatology, University of California, San Francisco, USA. [5]Department of Pediatrics, University of California, San Diego, USA. [6]Genomics Research Center, Academia Sinica, Taipei, Taiwan. [7]Institute of Cellular and Organismic Biology, Academia Sinica, Taipei, Taiwan. ✉e-mail: johnyu@gate.sinica.edu.tw

DSB stimulates cellular DNA repair mechanisms via error-prone non-homologous end joining (NHEJ) or homology-directed repair (HDR), which results in site-specific genetic alterations[15]. Therefore, DSB activates the NHEJ repair pathway leading to insertions or deletions at the target site, causing frameshift mutation in the coding sequence and gene knockout. Another use of CRISPR-Cas9 is to perform HDR to create point mutations and gene knock-in. Thus, this technology holds promise for correcting hereditary diseases caused by point mutations.

An important application of CRISPR-Cas9 is establishing genetically modified animal and cellular models of human diseases via gene knockout/ knock-in and site-specific mutagenesis[16,17]. This technology platform has progressed from being a research tool to one that promises clinical applications. While the utility of CRISPR-Cas9 genome editing for gene therapy in humans has been recognized and investigated[18,19], several studies demonstrated that Cas9-mediated genomic editing in human and mouse embryos led to on-target chromosome structure alterations and chromosome loss[20–23]. Other studies reported large DNA deletions and genomic rearrangements at the on-target sites in mouse ESCs, mouse hematopoietic progenitors, human differentiated retinal pigment epithelial cells, primary cells, and cancer cell lines[24,25]. Moreover, CRISPR-Cas9 edited fertilized zebrafish eggs showed structural variants (ranged from 4.8 Kb-deletions to 1.4 Kb-insertions) at on-target and predicted off-target sites with sequence similarity to the sgRNA and 26% of their offspring carry off-target mutations[26]. In addition, a recent study employing ultra-deep clinical next-generation sequencing in the coding region of 523 cancer-relevant genes demonstrated the safety of CRISPR/Cas9 genome editing in human hematopoietic stem and progenitor cells[27]. However, chromosomal aberrations outside the coding region of these genes were not surveyed. Since genome aberrations caused by CRISPR-Cas9 editing could lead to unforeseen pathogenic consequences, pre-testing for sequence variations is essential. In the literature, there are numerous methods used to analyze genomic alteration. Karyotyping is the oldest genetic method for chromosome alterations larger than 5 Mb; it can detect aneuploidy as well as transpositions, deletions, duplications, and inversions. Chromosomal microarray (CMA) has been used to determine chromosomal imbalances such as amplifications and deletions such as copy number variants (CNV). CMA provides submicroscopic resolution allowing us to visualize small regions that karyotyping cannot detect. Depending upon the particular array and how many DNA probes are used, it is possible to detect as small as 10 Kb. In contrast to microarray methods, next-generation sequencing (NGS), also known as high throughput sequencing, directly determines the nucleic acid sequence of a given DNA.

In this study, we investigate the whole genome integrity of thirteen CRISPR-Cas9 edited human pluripotent stem cell lines to uncover possible structural alterations, particularly those that may not be detected by short-read sequencing. We perform linked-read sequencing by 10x Genomics and optical genome mapping[28–30] by Bionano Genomics Saphyr System to examine the entire genome structure and sequence of the edited genomes. Here, we identify one on-target large structural variant (SV) (>50 Kb) and two large SVs at atypical non-homologous off-target sites, which were previously unrecognized as consequences of CRISPR-Cas9 mediated genome editing. Importantly, we illustrate a strategy for whole genomic analysis using linked-read sequencing and optical mapping to detect and validate CRISPR-Cas9 genome editing outcomes. This study may provide an approach for reducing the risk of unexpected adverse effects in research and clinical applications.

## Results

### Whole-genome sequencing of CRISPR-Cas9-edited human iPSC by linked-read sequencing
To evaluate the genome integrity after CRISPR-Cas9 mediated gene editing, we first generated *B2M* knockout iPSCs. The *B2M* gene on chromosome 15 of iPSC NC01 was edited by transient expression of

Cas9 nuclease and a sgRNA targeting the first exon of the *B2M* gene (Fig. 1a). Three single-cell clones (*B2M*[−/−]−1, −2, and −3) were randomly selected and isolated. The frameshift mutations were detected in exon 1 of the *B2M* gene and verified by Sanger sequencing (Fig. 1b). None of these three *B2M*[−/−] clones express HLA-class I molecules (HLA-ABC) on the cell surface, demonstrating the successful knockout of the *B2M* gene (Fig. 1c).

Next, we performed whole genome sequencing to examine the entire genome integrity of these single-cell clones of the CRISPR-Cas9-edited iPSCs. The short-read sequencing, which currently dominates genomic analysis, may identify some structural variants, but it cannot delineate repetitive regions nor resolve the human genome into haplotypes. In addition, the diploid nature of the human genome makes it even more difficult to detect large SVs in the heterozygous state. To overcome these issues, we performed linked-read sequencing to inspect the integrity of CRISPR-Cas9-edited genomes.

High molecular weight DNAs consisting of long DNA fragments, with 90–95% >20 Kb in length, were prepared and subjected to 10x Genomics Linked-Reads sequencing. Reads generated have an average mean depth of 52.8X, out of which an average of 95.4% of the reads could be mapped to the reference genome GRCh38 (Genome Reference Consortium Human Build 38), resulting in more than 99.1% of SNPs being phased (Supplementary Table 1).

### Knockout of the *B2M* gene by CRISPR-Cas9 genomic editing induces unexpected chromosomal large structural variants
The linked-read sequencing data was analyzed using the Long Ranger software (v2.2.2) pipeline with default options. The linked-reads were mapped to the human reference genome, GRCh38, using the Lariat. We used Long Ranger for structural variant detection and Loupe (v2.1.1) for visualization. As compared to the reference genome, there are many large SV calls found in the parental (NC01) and three *B2M* knockouts derived from NC01 (Fig. 2a). However, when compared to the NC01, there are two tentative large SV calls (marked as *) which were detected in *B2M*[−/−]−2, but not in the NC01 nor the other two knockouts (Fig. 2a).

To confirm these two large SVs found in *B2M*[−/−]−2, we examined the barcode overlapping of these two large SVs, plotted by Loupe software (10x Genomics). As shown in Fig. 2b, the large SV1 is a 136 Kb-heterozygous deletion on chromosome 3 (chr3:41537429-41673419). The large SV2 is a 68 Kb-heterozygous deletion on chromosome 15 (chr15:44710621-44778608). In Fig. 2c, the linked-reads assigned to each haplotype (Hap 1 or 2) of chromosome 3 and 15, respectively, are grouped and color-coded: a heterozygous deletion of approximately 136 Kb on Hap 2 of the chromosome 3 (large SV1) and a 68 Kb-heterozygous deletion on Hap 2 of chromosome 15 (large SV2) was found.

Optical genome mapping is another technology that provides structural information about a single long DNA molecule. It is powerful for examining structural variants due to the much longer length of optical genome mapping analysis (up to 2.5 Mb) compared to the sequencing reads[28,31]. Optical genome mapping was performed on the NC01 and three *B2M* knockouts to confirm the finding of large SVs identified by the linked-reads. As shown in Fig. 2d, optical genome mapping revealed a 136 Kb-heterozygous deletion on the chr3:41.5 Mb locus and a 68 Kb-heterozygous deletion on the chr15:44.7 Mb locus in the *B2M*[−/−]−2, consistent with the two large SVs detected by linked-reads. Likewise, optical genome mapping did not reveal any large SVs in the NC01, *B2M*[−/−]−1, and *B2M*[−/−]−3, consistent with the linked-read sequencing. These results thus strongly confirmed that the CRISPR-Cas9-mediated gene knockout induces large chromosomal SVs.

### Validation of large SVs using polymerase chain reaction and identification of an on-target and an unexpected off-target large SV
To further confirm the existence of two large SVs in *B2M*[−/−]−2 identified by linked-reads and optical genome mapping, we performed further

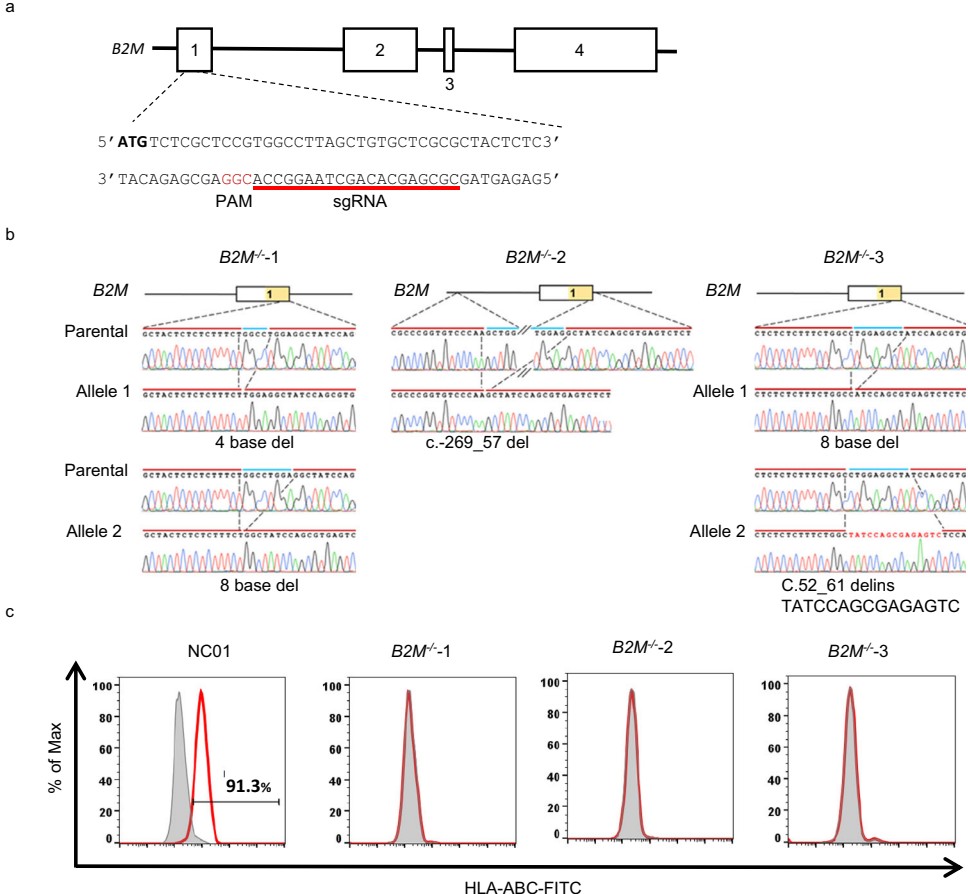

**Fig. 1 | The gene editing for B2M and workflow analysis. a** Schematic presentation of human *B2M* gene and sequence of *B2M* exon 1 showing PAM site and the genomic target for sgRNA. Boxes represent exons of the *B2M* gene. **b** The sequences and chromatogram of the frameshift mutations in *B2M^−/−* clones. The *B2M^−/−*−1, −2, and −3 single-cell clones were isolated after CRISPR-Cas9-mediated genome editing. The DNA mutations of each allele were demonstrated by sequencing analysis. **c** Knockout of *B2M* in iPSCs abrogates surface expression of HLA class I (HLA-ABC) detected via flow cytometry. The red color represents the expression of HLA-ABC, and the gray color represents the isotype control. *n* = 4 replicates. The gating strategies are provided as a Source Data file.

validation using polymerase chain reaction (PCR). The pair of primers, shown as a and c in chr3 and chr15, separately, in Fig. 3, were designed to cover enormously big genomic DNA fragments (136- and 68-Kb in size) that exceed the current PCR technology to generate any PCR amplicons. On the other hand, if large DNA deletion did occur in these regions, PCR amplicons of much shorter DNA segments would be generated. Genomic DNA prepared from the *B2M^−/−*−2 clone was amplified using these two pairs of primers, and two specific PCR amplicons of 1.1-Kb and 1.3-Kb in size were found (Fig. 3, a+c). In addition, the PCR products were verified by Sanger sequencing, and the break junctions were confirmed. These results were consistent with the notion that this clone had two large structural variants, SV1 and SV2, on chromosomes 3 and 15.

In contrast, PCR amplification of DNA from NC01, *B2M^−/−*−1, or *B2M^−/−*−3 using the same pair of primers did not generate any specific PCR products (Fig. 3, a+c). To rule out the possibility that this lack of PCR amplicons in NC01, *B2M^−/−*−1, and 3 were attributed to inadequate quality of genomic DNA preparation, we designed a second pair of primers, a and b, shown in Fig. 3, which targeted short genomic regions. With these pairs of primers, PCR amplicons with approximately 0.75 Kb for the large SV1 region on chr3 and 1.1 Kb for the large SV2 region on chr15 were detected in all DNA samples obtained from NC01 and three knockouts (Fig. 3, a+b). Therefore, the generation of PCR amplicons from the intact genomic region of one allele and the deletion region of the other allele from *B2M^−/−*−2 indicates that the

deletions are heterozygous, also consistent with the findings by linked-reads and optical genome mapping.

Additionally, the large SV2 on chr15 (chr15:44710621-44778608) is an on-target DNA deletion, which may have derived from the DNA repair of double-strand break caused by sgRNA/Cas9 targeting where the sgRNA is located at chr15:44711556-44711575. The on-target large DNA deletion has been reported for gene knockout mediated by CRISPR-Cas9 genomic editing[24].

To assess the possibility of predicted off-target effect on large SV1, we used the Cas-OFFinder[32] algorithm to predict the potential off-target sites close to the chromosome region of large SV1 with mismatch numbers equal to or less than six and DNA bulge size equal to or less than two. Surprisingly, no predicted off-target site close to large SV1 was found, as shown in Supplementary Table 2. In addition, we used CRISTA[33], an off-target search tool that implements many features, such as GC contents, RNA secondary structure, DNA methylation, epigenetic factors, and Elevation[34], that takes both sequence and chromatin accessibility features to predict the potential target sites. However, as shown in Supplementary Tables 3 and 4, these predicted potential target sites are too far from the large SV to account for Cas9-mediated DNA cleavage. Besides, our whole genome analysis did not reveal any sequence/structural abnormalities in these predicted off-target sites. Accordingly, the large SV1 may be induced by an atypical non-homologous off-target event without sequence similarity to the sgRNA. Such an unexpected, atypical non-homologous off-target large

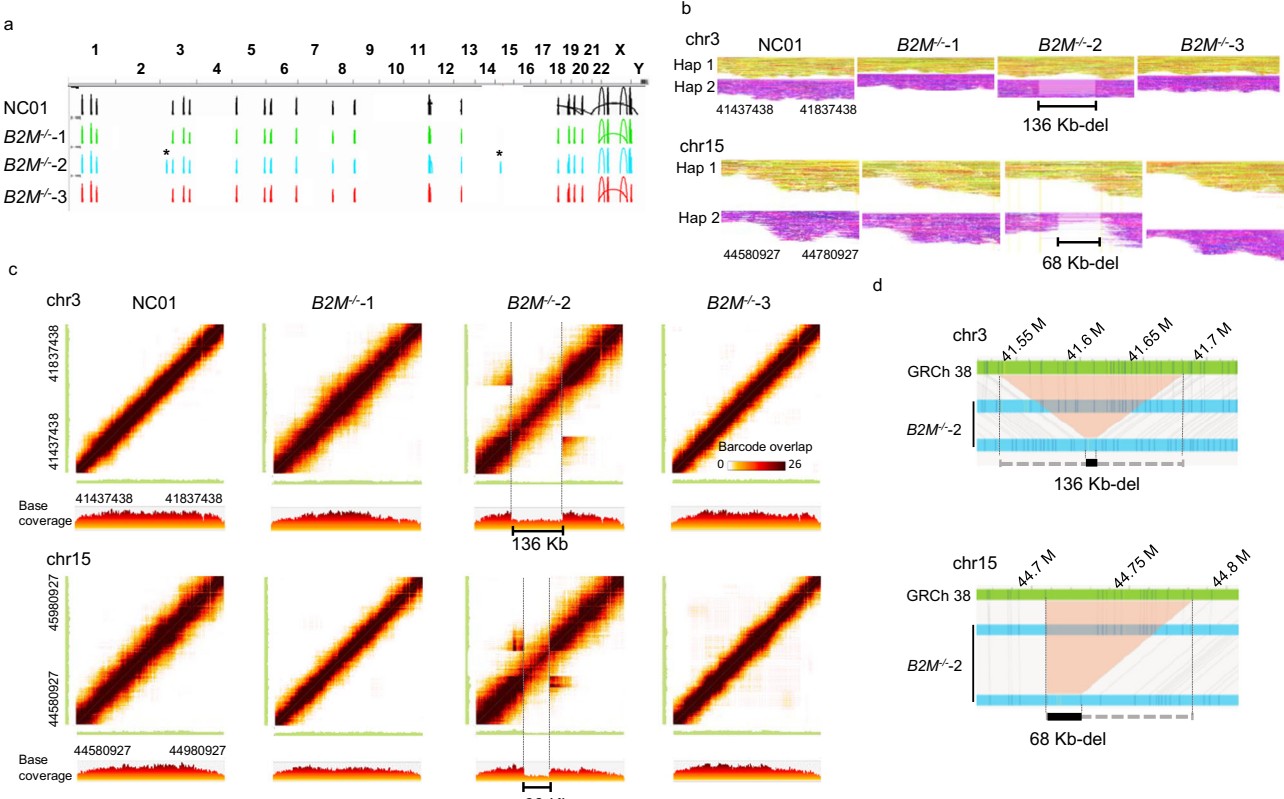

**Fig. 2 | Large SVs arise during the CRISPR-Cas9-mediated gene knockout of B2M. a** The large SV calls are constructed from linked-reads of the parental (NC01) and three single-cell clones of *B2M* knockouts. Peaks represent the predicted large SV calls compared to GRCh38. The asterisks * indicated large SVs on chromosomes 3 and 15 in *B2M*[-/-]−2, which were not detected in the NC01 and *B2M*[-/-]−2 and 3. **b** Matrix view of the overlapping barcodes analyzed with Loupe software (10x Genomics) showed heterozygous deletions in *B2M*[-/-]−2 on chr3:41437438−41837438 and chr15:44580927−44980927, respectively. The matrix view was plotted with the dark brown color representing the shared barcodes between two genomic segments marked on the *X*- and *Y*-axis. The *X* and *Y* axes correspond to the same genome region, so the barcode overlap matrix is symmetric. The diagonal shows the number of barcodes in each position along the displayed region. The colored band around the diagonal reflects long molecules that span several kilobases, thus

generating barcode overlaps across their span. The color intensity drops suggest a relative drop in the number of molecules in that region. Therefore, the drop in coverage and the off-diagonal barcode overlap suggest a heterozygous deletion. The linear view represents the base coverage along the *X*-axis segment. The 136 and 68 Kb indicate heterozygous deletions. **c** The phased reads graphs showed a 136 Kb-heterozygous deletion on chr3:41437438−41837438 and a 68 Kb-heterozygous deletion on chr15:44580927−44780927 in *B2M*[-/-]−2. Reads are partitioned into distinct haplotypes 1 (Hap1) and 2 (Hap2). **d** The optical genome mapping revealed a 136 Kb-heterozygous deletion on the chr3:41.5 Mb locus and a 68 Kb-heterozygous deletion on chr15:44.7 Mb locus in the *B2M*[-/-]−2. The gray lines indicate the alignment between the reference (GRCh38; green) and the assembled maps (*B2M*[-/-]−2; blue). The light red area indicates the deletions.

SV is, to the best of our knowledge, the first discovery in the CRISPR-Cas9 edited genomes, which implies the emergence of unforeseen pathological consequences during CRISPR-Cas9 mediated gene knockout.

## The gene knock-in for *APP* c.2033G>C point mutation by CRISPR-Cas9 editing also induces an atypical non-homologous off-target large SV

To investigate whether the observed atypical non-homologous off-target large SV may be incurred by CRISPR-Cas9 mediated gene knock-in, we performed CRISPR-Cas9 genome editing to generate point mutation for *APP* c.2033G>C in human iPSC-71 as illustrated in Fig. 4a. A homozygous gene-mutated-(*APP*[C/C]) and a heterozygous mutated (*APP*[C/G]) single-cell clones were isolated. The base substitution was verified using Sanger sequencing. In addition, whole-genome linked-read sequencing (Supplementary Table 5) and optical genome mapping were used to examine the genome integrity after CRISPR-Cas9 editing. Compared to the reference genome, there are several large SV calls predicted in parental (iPSC-71) and *APP* knock-in genomes (Fig. 4b). Moreover, there is a tentative large SV call (marked as *), which was presented only in *APP*[C/C], but not in the iPSC-71 nor *APP*[C/G] (Fig. 4b). Based on the linked-reads (Fig. 4c, d) and optical genome mapping (Fig. 4e), this large SV

was found to consist of a 91.2 Kb-heterozygous deletion on chromosome 3. The breakpoint junction of this 91.2 Kb-heterozygous deletion was confirmed by PCR which showed the generation of 1.3 kb PCR product in the *APP*[C/C] (Fig. 4f) but not in the iPSC-71 nor *APP*[C/G]. Besides, the sgRNA/Cas9 targeting locus is on chr21:25897598-25897615, whereas the large SV locus is on chr3:39882164−39973392 without any off-target site as predicted by Cas-OFFinder, CRISTA, and Elevation (Supplementary Tables 6−8). Thus, these results suggest that the 91.2 Kb-heterozygous deletion observed in the *APP*[C/C] is another unexpected, atypical non-homologous off-target large SV induced by CRISPR-Cas9-mediated gene knock-in. Furthermore, we performed CIRCLE-seq[35,36] to evaluate the specificity of sgRNA used in *B2M* knockout and *APP* knock-in (Supplementary Fig. 1). It was observed that there were efficiently targeted DNA cleavage sites (i.e. high CIRCLE-seq read counts) of the on-target *B2M* in chr15 (denoted by an asterisk, Supplementary Fig. 1a) and *APP* in chr21 (denoted by an asterisk, Supplementary Fig. 1b); but no reads detected on the chr3, where the two large SVs were detected in our studies. Therefore, these results further support our conclusion that the large SVs are independent of the homologous targeting by sgRNA.

In addition, we performed optical genome mapping to study the occurrence of SVs in single-cell clones isolated from the PiggyBac

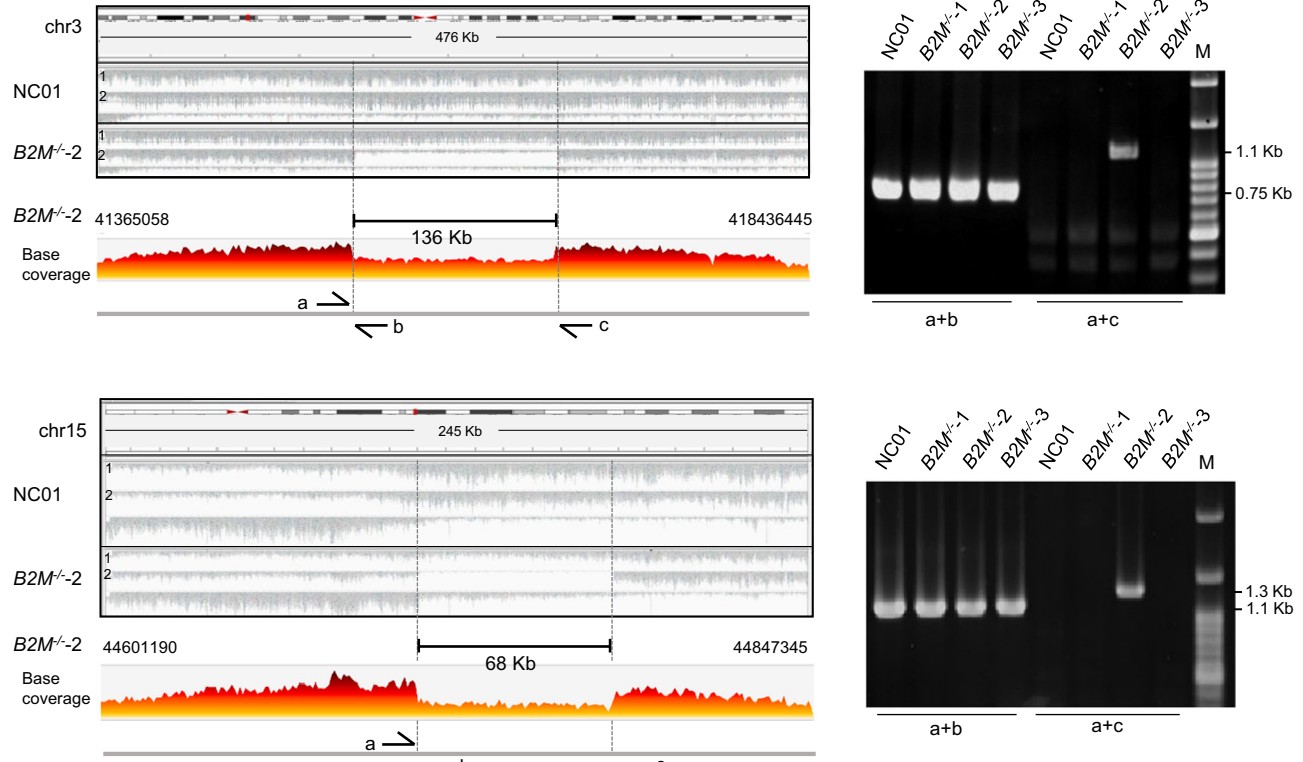

**Fig. 3 | PCR validation for the heterozygous large SVs in *B2M*[-/-]−2.** The phased BAM format was performed with the Integrative Genomics Viewer (IGV_2.7.2)[48–50] sorted with 10x Genomics linked-read haplotype tag (HP; haplotype of the molecule that generated the read). The region of large SVs in the parental (NC01) and *B2M*[-/-]−2 can be phased into two strands with an HP tag (left panel). Agarose gel electrophoresis images of the PCR products amplified from NC01 genomic DNA and *B2M* knockouts (right panel). Primers a and b are designed to target the breakpoint junctions and primers a and b for the intact genomic region. The base coverage of large SVs in *B2M*[-/-]−2 was visualized by Loupe. M is the size marker. Primers are listed in Supplementary Table 12. *n* = 3 replicates for PCR analysis. Source data are provided as a Source Data file.

transposon-mediated knock-in in the two parental iPSC lines, which were found to have large SVs in the Cas9-mediated gene editing: *ETV2i2* knock-in in NC01 (the parental cell line of *B2M*ko) and *NGN2* knock-in in iPSC-71 (the parental cell of *APP*[C/C] knock-in). The average N50 of those molecules was 246 Kb (range: 225-275 Kb), the average mapping ratio mapped to reference genome from molecules was 81.5% (range: 78.7-83.5%), and the average effective coverage was 84.8X (range: 57.1−111.48X). Several LSVs were detected in parental (NC01) and *ETV2i2* knock-in genomes compared to the reference genome (GRCh38). However, none of these large SVs was found only in the *ETV2i2* gene knock-in but not in the parental genomes (NC01; Supplementary Fig. 2a). Similarly, we could not find any large SVs which are present only in the *NGN2* knock-in genomes but not in the parental iPSC-71 (Supplementary Fig. 2b) either. These results indicate that no unexpected large SVs occurred in the genomes of PiggyBac transposon-mediated gene knock-in clones derived from these two iPSCs (NC01 and iPSC-71). Therefore, these findings suggest that the atypical non-homologous off-target large SVs detected in the *B2M*[-/-]−2 and *APP*[C/C] clones may not have been caused by the cloning, prolonged expansion, and proliferation of these two iPSC lines and that the unexpected large SVs detected were not simply artifacts of the cell lines used in this study.

Moreover, we also investigated genome changes in *B3GALT5* knockout[37] in human embryonic stem cell H9, *LRRK2* (G2019S) knock-in[38] in iPSC ND40019*C, and *DSG2* (F531C) gene knock-in in iPSC-71 (Supplementary Table 9). As shown in Supplementary Fig. 3, no large SV was revealed in these three edited genomes. In this study, two of the five edited cases of thirteen human pluripotent stem cell lines contained atypical non-homologous off-target large SVs incurred during CRISPR-Cas9-mediated genome editing.

## Discussion

Since genomic editing has clinical therapeutic implications for genetic diseases, it is crucial to understand the potential genome changes associated with CRISPR-Cas9 genome editing. This study combined the 10x linked-read sequencing and optical genome mapping to identify the large chromosomal SVs in CRISPR-Cas9 edited genomes. Through these two approaches, we uncovered a large SV at the on-target site, an atypical non-homologous off-target large SV in a *B2M*[-/-] knockout clone, and another atypical non-homologous off-target large SV in *APP*[C/C] knock-in clone. However, no large SV was found in *B3GALT5* knockout, *LRRK2* (G2019S) knock-in, and *DSG2* (F531C) gene knock-in genomes. Two of the 13 edited genomes we examined carried a large SV at an unpredicted atypical non-homologous off-target site; one of these two genomes had an additional large SV at the on-target site, which is expected. Altogether, in our CRISPR-Cas9 edited genomes, ~15% acquired unexpected, atypical non-homologous off-target large SVs.

Intriguingly, the two large SVs identified in these independently derived CRISPR-edited iPSC clones are located in chromosome 3p, which is distinctly apart from the chromosome 15 (*B2M*) and 21 (*APP*), where the target genes reside. While Chromosome 3 spans about 198 Mb, the two large SV identified are ~1.65 Mb apart. Whether the relative proximity of these large SV on two unrelated CRISPR-edited iPSC clones is purely co-incidental or related to some unknown mechanisms that predispose this region of chromosome 3 to genetic alterations awaits further studies. Furthermore, we browsed the chromosomal fragile sites on HumCFS, a human chromosomal fragile sites database[39]. There are four known chromosomal fragile sites on the chr3: FRA3A (chr3:23900001-26400000), FRA3B (chr3:58600001-63700000), FRA3C (chr3:182700001-187900000), and FRA3D

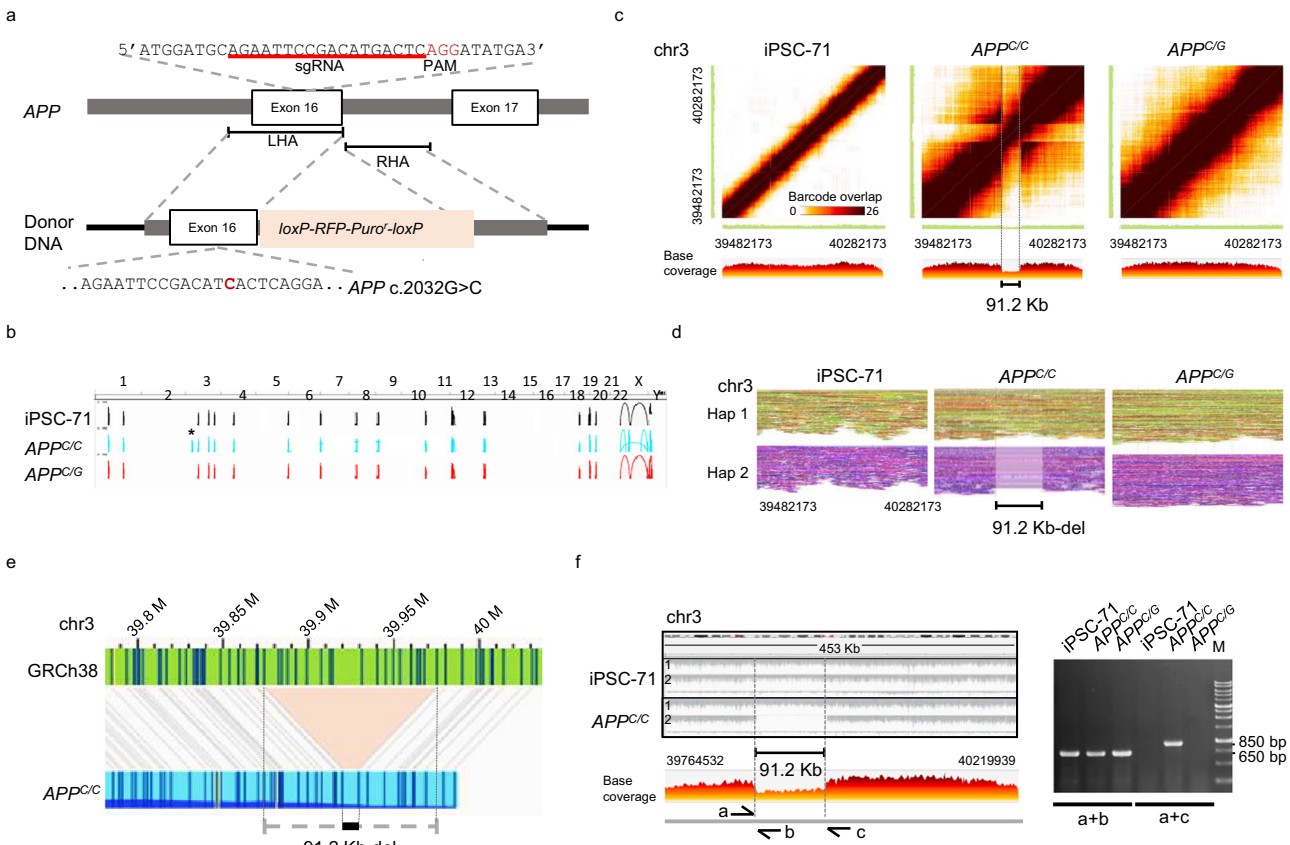

**Fig. 4 | Large SV was found after CRISPR-Cas9 mediated gene knock-in. a** The strategy for the base substitution of exon 16 of *APP* gene in human iPSC-71 by CRISPR-Cas9 mediated gene knock-in. The schematic presentation and the sequence of *APP* exon 16 loci with PAM, the targeting site for the sgRNA, and the donor DNA used for homology-directed repair (HDR) are shown. The *loxP-RFP-Puro^r-loxP* cassette was flanked by two homology arms. LHA, left homology arm (633 bp); RHA, right homology arm (582 bp). Boxes represent exons. **b** Large SV calls were constructed from linked-reads of the genome from iPSC-71, *APP^{C/C}*, and *APP^{C/G}*. The asterisk * indicates a large SV found in the *APP^{C/C}* but not in the parental (iPSC-71) nor *APP^{C/G}*. **c** Heat map of the overlapping barcodes in chr3:39482173–40282173 showing a heterozygous deletion in *APP^{C/C}*. The matrix view was plotted by the Loupe, where the dark brown color represents the shared barcodes between the two genomic segments on the X- and Y-axis. The linear view represents the base coverage along the X-axis segment. **d** The phased reads graph indicates a 91.2 Kb-heterozygous deletion on chr3 in the *APP^{C/C}* genome. Reads were partitioned into distinct haplotypes: haplotype1 (Hap1) and haplotype 2 (Hap2). **e** The optical genome mapping reveals a 91.2 Kb-heterozygous deletion (black bar) in chr3 of the *APP^{C/C}*. The gray lines indicate the alignment between the reference (GRCh38; green) and assembled map (*APP^{C/C}*; blue). The light red area shows the deletion. (**f**) PCR confirmation of the heterozygous deletion. PCR verification with primers a and b and using iPSC-71, *APP^{C/C}*, and *APP^{C/G}* genomic DNA as the template. The primer pair, a and b, were designed to amplify breakpoints of the deletion, which produce a unique amplicon in the mutant strand harboring the large SV; and the primer pair, a and b, were for the wild-type strand. The region of the large SV in the iPSC-71 and *APP^{C/C}* was phased into two strands with an HP tag (left panel) and base coverage of *APP^{C/C}* visualized by Loupe. M is the size marker. Primers are listed in Supplementary Table 12. n = 3 replicates for PCR analysis. Source data are provided as a Source Data file.

(chr3:148900001-160700000). However, the large SV in *B2M^{-/-}*−2 is at chr3:41535938-41680677, and the large SV in *APP^{C/C}* is at chr3:39882164−39973392, neither of which are fragile sites. Therefore, the large SVs reported in this study are not in the known chromosomal fragile sites. The large SV detected in the *B2M^{-/-}*−2 clone contains the *ULK4* gene; ULK4 belongs to the family of unc-51-like serine/threonine kinases, which participates in a conserved pathway involving both endocytosis and axon growth[40–42]. Sequence variations in this gene have been associated with schizophrenia and bipolar disorder[43]. The large SV detected in the *APP^{C/C}* contains the *MYRIP* (Myosin VIIA and Rab interacting protein) gene, which is predicted to exhibit binding activity for actin and myosin, to be involved in positive regulation of insulin secretion and functions as a component of the exocytosis machinery[44,45]. Thus, neither ULK4 nor MYRIP is known to be directly involved in cell proliferation, cell death, and cell signaling, which might be predicted to give the proliferative advantage to the clones.

The linked-read sequencing (up to 150 Kb) provides a clear advantage over short reads (typically 150–300 bp) alone, allowing for the construction of long-range haplotypes and promising better long-

range contiguity and resolution of repetitive regions. Alternatively, one of our authors, Dr. Pui-Yan Kwok, works on another alternative by combining high-accuracy short reads and PacBio-CLR (the continuous long read) to resolve structural variations and provide haplotype phasing. Other options include other long read sequencings such as PacBio and Nanopore sequencings, but these two long read technologies have their own problems, such as sequence accuracy and high cost. In contrast to DNA sequencing read, optical genome mapping produces single long molecule (~225 Kb) maps which could cover heterozygous genome regions with different haplotypes that cannot be easily spanned by sequencing read. On the other hand, optical genome mapping detects SVs across the whole genome but does not provide sequence-level information or precise SV breakpoints. Therefore, combining linked-read sequencing and optical genome mapping would be a better option for genome integrity analysis.

A significant issue of CRISPR-Cas9 technology is the unintended mutations for cell replacement therapy at the genome locus other than the targeted site. Such off-target mutations can have profound consequences as they might disrupt the function or regulation of the

non-targeted genes. In addition, large SVs recognized in this study, occurring at either on-target or atypical non-homologous off-target sites, could pose further concerns. Such DNA lesions may lead to pathological or carcinogenic changes in stem cells, which have a long replicative lifespan and may become non-functional or neoplastic with time. Therefore, the atypical non-homologous off-target large SVs reported here illustrate a need to thoroughly examine the genome integrity when CRISPR-Cas9 editing is conducted. Besides, the plasmid DNA delivery was used in this study to account for CRISPR-Cas9 editing, and we wondered if ribonucleoprotein (RNP) delivery could see whether the atypical non-homologous off-target large SVs would happen. A recent paper performed the genome-editing simultaneously of *HLA-A* and *-B* genes located on chr6 with the RNP delivery system[46]. In the original report, chromosome karyotyping revealed that one of the 15 selected iPSC clones had translocation of chr2 distal to 2p16−22, to the terminal end of the long arm of chr15. Therefore, we employed the Cas-OFFinder and CRISTA algorithms to examine the potential off-target sites close to the chromosome region. As it turned out, we did not detect potential off-target sites close to this translocation region (Supplementary Tables 10 and 11). These results thus suggest that this large chromosome translocation detected by karyotyping is also an unexpected, atypical non-homologous off-target large SV induced by CRISPR-Cas9 using RNP delivery. However, the original paper did not mention that this off-target large SV site, without sequence similarity to the sgRNA, occurred during RNP genome editing. In addition, the finding of the translocation of this big chromosomal fragment was detected by chromosome karyotyping, which has the limitation for detecting rearrangements with less than 5 Mb of DNA, although the use of multicolor fluorescence in situ hybridization can improve the resolution to ~100 Kb−1 Mb in size. Improving the ability to detect such un-intended large SVs to ensure genome integrity will maximize the safety and chance for clinical implementation of site-specific genome editing therapies.

In conclusion, gene editing of the same iPSC line with CRISPR-Cas9 does not always induce large SV at atypical non-homologous off-target sites in all gene-edited clones. For instance, the large SV was only observed in one of three *B2M*ko clones. Thus, we do not advocate stopping the use of this powerful genome editing tool based on our findings. Instead, we propose a strategy for detecting and validating CRISPR-Cas9 genome-editing outcomes using linked-reads and optical genome mapping. Our strategy of whole genomic analysis using linked-read sequencing and optical genome mapping may provide a valuable strategy for confirming the genome integrity of the CRISPR-Cas9 edited clones before clonal expansion and long-term ex vivo culture for research and clinical applications.

## Methods

### Cell culture and CRISPR-Cas9-mediated genome editing
Human iPSCs were maintained with Matrigel (human embryonic stem cell–qualified matrix, Corning, No.354277) in Essential 8™ Medium (E8, Gibco™, No. A1517001). The medium was changed every 24 hours, and cells were passaged after 5 min incubation with 0.5 mM EDTA (Invitrogen™, No.15575020) at room temperature. For replating iPSCs, the E8 medium was supplemented with a 2 μM ROCK inhibitor, thiazovivin (Selleckchem, No. S1459).

For gene knockout, $2 \times 10^6$ single-cell suspensions of human iPSCs were electroporated with 1 μg of the Cas9 and gRNA containing plasmid DNA using the Neon Transfection System (Invitrogen) following the settings of 1500 V, 20 ms pulse width, and a single pulse. For gene knock-in, $2 \times 10^6$ human iPSCs were electroporated with 1 μg of the Cas9 and gRNA-containing plasmid DNA and 1 μg of the donor DNA using the settings of 1500 V, 20 ms pulse width, and a single pulse. One day later, the medium was changed into E8 medium, supplemented with 1 μg/ml puromycin for 48 hours. The puromycin-resistant iPSCs were plated at a low density in 6-well

plates and cultured for days until the surviving colonies derived from a single cell were manually picked. After clonal expansion, the genomic DNA of these candidate clones was isolated and used for PCR and sequencing. The cell lines used in this study are H9 (NSC-H9, WiCell), NC01 (generated in our lab), iPSC-71 (material transfer from Joseph C. Wu, Stanford Cardiovascular Institute), and ND40019*C[38]. The sgRNA sequences are listed in Supplementary Table 12.

### Flow cytometry analysis
For surface staining, $5 \times 10^5$ cells were incubated with antibodies in total 100 μl at 4 °C for 30 min and analyzed with SA3800 Spectral Analyzer (SONY). The antibodies were HLA-ABC monoclonal antibody (W6/32, FITC, eBioscience, 11-9983-42, 1:200) and mouse IgG2a kappa Isotype Control (eBM2a, FITC, eBioscience™, 11-4724-42, 1:200).

### High molecular weight DNA extraction and 10x Genomics sequencing and assembly pipeline
High-molecular-weight genomic DNA (HMW gDNA) extraction, sample indexing, and partition barcoded libraries were done according to 10x Genomics (Pleasanton, CA, USA), Chromium Genome User Guide. The HMW gDNA was extracted from each sample with the Bionano Prep™ kit (Bionano Genomics). Then HMW gDNAs were subjected to 10x Genomics Linked-Reads sequencing on the NovaSeq 6000 Sequencing System (Illumina) to 60x read depth.

Long Ranger (v2.2.2) was used for analyzing the 10x sequencing data with the "WGS" pipeline and the default settings. The sequence files were processed and aligned with GRCh38 via Long Ranger BASIC and ALIGN Pipelines (10x Genomics) for linked-read alignment, variant calling, phasing, and structural variant calling. The Lariat aligner mapped the linked-reads to the reference genome (GRCh38/hg38). After variant calling, a phasing method optimized for a 10x barcode was used to phase the identified SNPs. The output of the Long Ranger pipeline was analyzed with the Loupe (v2.1.1) to visualize large SVs, inter-chromosomal translocations, gene fusions, and inversions or deletions. Loupe viewer also displayed the analyzed genome with two haplotypes.

### Bionano whole-genome mapping
The HMW gDNAs were prepared for Bionano Genomics optical genome mapping library following the Bionano Prep Direct Label and Stain (DLS) protocol. Briefly, cells were embedded into low-melting-point agarose gel plugs (BioRad #170-3592, Hercules, CA, USA). Then, plugs were incubated with lysis buffer and proteinase K overnight at 50 °C, solubilized with agarose (Thermo Fisher Scientific) at 50 °C for 40 min, and the purified DNA was subjected to drop-dialysis for one h. Next, the HMW gDNA was subjected to an enzymatic labeling approach for direct fluorescent labeling by the Direct Label Enzyme (DLE-1). After sequence-specific labeling with DLE-1, the DLE libraries were applied to optical genome mapping on the Bionano Genomics Saphyr System to 60x coverage. Next, single-molecule maps were assembled de novo into genome maps using the assembly pipeline developed by the Bionoano Solve pipeline (v3.3, v3.6.1 and v3.7.01) with default setting[47]. All structural variations (SVs), including deletions, insertions, inversions, and translocations, were annotated to GRCh38 by the Variant Annotation Pipeline of Bionoano Solve (v3.3, v3.6.1, and v3.7.01). The SVs of interested regions were shown after being compared to the GRCh38.

### sgRNA off-target analysis
The Cas-OFFinder program (http://www.rgenome.net/cas-offinder/, with mismatch numbers equal to or less than six and DNA bulge size equal to or less than two), CRISTA (https://crista.tau.ac.il/), and Elevation (https://crispr.ml/) were used to predict the off-target sites for sgRNAs used in this study.

## CIRCLE-seq library preparation

CIRCLE-seq experiments were performed on genomic DNA from the same cells in which they were used for genome editing (either NC01 or iPSC-71 cells). CIRCLE-seq library construction followed the protocol published in *Nat Protoc. 2018*[35]. Briefly, purified genomic DNA was sheared to an average length of 300 bp and circularized at low DNA concentration. In vitro cleavage reactions were performed with Cas9 nuclease buffer (NEB, B6003S), 90 nM SpCas9 protein (NEB, M0386M), 90 nM synthesized gRNA, and 250 ng of circularized DNA. Digested DNA products were A-tailed, ligated with a hairpin adaptor, treated with USER enzyme (NEB, M5505L), and amplified by PCR using KAPA HiFi polymerase (Roche). The libraries were sequenced with 150 bp paired-end reads on the MiSeq Sequencing System (Illumina).

## Statistics and reproducibility

No statistical method was used to predetermine the sample size. No data were excluded from the analyses. The experiments were not randomized. The investigators were not blinded to allocation during experiments and outcome assessment.

## Reporting summary

Further information on research design is available in the Nature Portfolio Reporting Summary linked to this article.

## Data availability

The 10x Genomics sequencing data of genomes generated in this study have been deposited in the NCBI SRA database under accession code PRJNA943092. Source data are provided with this paper.

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

## Acknowledgements
This study was supported by grants to J. Yu from Chang Gung Medical Foundation in Taiwan OMRPG3C0048 and the National Science and Technology Council, Taiwan NMRPD1M0092. In addition, we thank Professor Akitsu Hotta from CiRA, Kyoto University, for the discussion and advice on RNP delivery. Finally, we thank the National Center for Genome Medicine in Taiwan and Feng-Jen Hsieh for technical support.

## Author contributions
H.H.T., A.L.Y., and J.Y. developed the study concept design and writing of the manuscript. H.H.T., H.J.K., M.W.K., C.H.L., C.M.C., Y.Y.C., and H.H. Chen performed the experimental studies. H.H.T. and H.J.K. carried out the analysis. P.Y.K., A.L.Y., and J.Y. supervised the work.

## Competing interests
The authors declare no competing interests.
