## [Peer Review File · Nature Communications]

Reviewers' Comments:

Reviewer #1:

Remarks to the Author:

The manuscript by Liu et al reported that CRISPR-Cas9 induces large structural variants in three out of 11 human pluripotent stem cell lines, a topic of great importance for clinical CRISPR-Cas9 use. Interestingly, two large chromosomal deletions were observed at off-target sites without sequence similarity to the sgRNA. The authors investigate the whole genome integrity with two complementary methods very well chosen to analyze large genome deletions, a linked-read sequencing by 10X Genomics and an optical mapping by Bionano Genomics Saphir system. The manuscript is well written and easy reading it.

Major comments

1° The main originality of the manuscript is based on the atypical off-target mechanism of deletions, which would be independent of homologous recognition. These discoveries are important but surprising and to my mind need to strengthen the evidence of this non homologous dependent mechanism of deletion. This observation is based on the Cas-OFFinder algorithm which did not predict potential off-target sites close to the deleted regions. We agree that Cas-OFFinder is one of the best method to detect potential off-target by aligning gRNA sequences to the reference genome based on sequence homology. However, some off-target sites cannot be predicted solely based on sequence homology. Features that influence the nonspecific binding of CRISPR gRNAs need to be considered to increase the accuracy of off-target detection. We recommend to complete Cas-OFFinder analysis by the use of another software for the two targets (B2M and APP genes) : for example CRISPR Target Assessment (CRISTA) software implements multiple features (PAM type, nucleotide composition, GC content, chromatin structure, DNA methylation, RNA secondary structure, etc.) to predict cleavage propensity. Alternatively, Elevation software that takes both sequence and chromatin accessibility feature into consideration should be interesting.

2° in silico predictions should be confirmed by at least one in vitro additional technique such as CIRCLE-seq, a highly sensitive off-target detection method, or alternatively GUIDE-seq which is a good choice for measuring off-target specificity ex vivo in a cellular context.

3° To further increase the evidence of a non-homologous off-target effect, the use of high fidelity cas9 would be optimal for the two targets. Indeed, the persistence of an off-target deletion after transfection of a high-fidelity cas9 would be very convincing to support the very interesting but atypical hypothesis of the authors.

Minor

If the three proposed experiments confirm the non-homologous off-target mechanism, it would be important in the discussion to propose mechanistic hypotheses.

Reviewer #2:

Remarks to the Author:

Tsai et al in "Comprehensive genomic analysis reveals atypical non-homologous off-target large structural variants induced by CRISPR-Cas9-mediated genome editing" describe their work using whole genome sequencing and optical mapping to determine if CRISPR/Cas9 editing creates structural variants in iPSC lines. The data and conclusions are clearly presented but the findings are expected based on the prior literature and thus the impact of the work is lessened. It is already well known that one must determine the genomic integrity of both engineered and un-engineered iPSC lines over time because of the risk of structural variants. This focused study confirms that need.

Specifically, I have the following concerns:

1. The term "comprehensive" overstates their work. They did not do RNA expression analysis nor chromatin analysis, as just two examples that the analysis was not comprehensive. They simply did whole genome sequencing.
2. Would the variants have been identified by using simpler and cheaper SNP microarrays or array CGH?
3. The use of plasmid DNA to deliver the CRISPR-Cas9 is not state of the art. It is well known that

the probability of generating off-target changes is increased with prolonged nuclease expression. the authors should use RNP delivery and see if the same thing happens.

Reviewer #3:

Remarks to the Author:

Summary: Authors describe a methodology to detect on and off target genome changes using novel approach of combining optical genome mapping with linked read sequencing. this combination has been shown to be a comprehensive approach for detection of all classes of genome variants. The application of assessing genomic integrity in IPS cells is very important and pertinent today as cell therapies continue to make their way through drug development pipelines but without a consensus regarding how to assure the absence of deleterious genome changes. In addition to describing the methodology, the authors also reported on unexpected large structural variants that presumably occurred as a result of CRISPR/CAS genome editing. These large SVs occurred in 2/11 samples bringing into question the safety of the current genome editing approach and underlining the critical need to standardize genomic QC in cell therapy applications.

Review overview: this manuscript presents the assessment of 11 edited cell lines and presents a case for the need to have better tools for genome assessment after editing. This is a topic of very high interest as there is significant debate regarding best practices and the need of higher resolution is gaining realization.

The study size is moderate but there very little published to date relating to SVs and edited cells and the findings that 18% of clones harbor a presumably editing induced large SV is quite striking and should have a high profile readership.

Some considerations that deserve attention are: linked read sequencing via 10X genomics is not commercially available any longer. Is there a viable substitute or alternative method?

It would be good to discuss the complementarity of OGM and linked read sequencing for this application. In these results, both techniques found all reported variants so they provide value by orthogonal confirmation but what other complementarity is there? Could this work be done with just one technology or in a tiered fashion if there were time or price sensitivity, for example?

other points through the manuscript:

Intro - suggest to describe the standard methods for genome assessment (KT, CMA, NGS)

line 86: this reference is not related to Bionano but is a predecessor tech, suggest to add nanochannel (bionano) and FGA references: Lam et al, Shieh et al?

line 133: "optical genome mapping, (OGM)" is more commonly used. suggest to change reference to Lam et al 2012

line 135: "due to much longer DNA molecule length used for analysis (up to 2.5 Mbp)"

line 135: references are based on much older technology and 2 of them don't involve nanochannels. suggest: Lam et al, Mak et al, Chaisson et al or others related to SV detection by OGM using the Bionano platforms

Line 158 should be "a" and "b"

line 214: specify whether "additional large sv at target site" is expected or not.

Figure 2 should be described better, these visuals are not well known and some panels, especially panel b and c might not be well understood.

Supp figure 1: should be described better in the legend. Reader does not know what they are looking at or how they relate to SVs

Responses to the comments of the reviewers

As you will see from the reports copied below, the reviewers raise important concerns. We find that these concerns limit the strength of the study, and therefore we ask you to address them with additional work. Without substantial revisions, we will be unlikely to send the paper back to review. In particular, we ask that you perform the 1) RNP delivery, 2) extra off-target analysis, and 3) show that this can't be achieved with other methods, in line with the reviewer requests.

If you feel that you are able to comprehensively address the reviewers' concerns, please provide a point-by-point response to these comments along with your revision. Please show all changes in the manuscript text file with track changes or colour highlighting. If you are unable to address specific reviewer requests or find any points invalid, please explain why in the point-by-point response.

Reviewer #1 (Remarks to the Author):

The manuscript by Liu et al reported that CRISPR-Cas9 induces large structural variants in three out of 11 human pluripotent stem cell lines, a topic of great importance for clinical CRISPR-Cas9 use. Interestingly, two large chromosomal deletions were observed at off-target sites without sequence similarity to the sgRNA. The authors investigate the whole genome integrity with two complementary methods very well chosen to analyze large genome deletions, a linked-read sequencing by 10X Genomics and an optical mapping by Bionano Genomics Saphir system. The manuscript is well written and easy reading it.

Major comments

1° The main originality of the manuscript is based on the atypical off-target mechanism of deletions, which would be independent of homologous recognition. These discoveries are important but surprising and to my mind need to strengthen the evidence of this non homologous dependent mechanism of deletion. This observation is based on the Cas-OFFinder algorithm which did not predict potential off-target sites close to the deleted regions. We agree that Cas-OFFinder is one of the best methods to detect potential off-target by aligning gRNA sequences to the reference genome based on sequence homology. However, some off-target sites cannot be predicted solely based on sequence homology. Features that influence the nonspecific binding of CRISPR gRNAs need to be considered to increase the accuracy of off-target detection. We recommend to complete Cas-OFFinder analysis by the use

of another software for the two targets (B2M and APP genes) : for example CRISPR Target Assessment (CRISTA) software implements multiple features (PAM type, nucleotide composition, GC content, chromatin structure, DNA methylation, RNA secondary structure, etc.) to predict cleavage propensity. Alternatively, Elevation software that takes both sequence and chromatin accessibility feature into consideration should be interesting.

We appreciate the comment from the reviewer that “off-target sites cannot be predicted solely based on sequence homology”. As suggested, we used “CRISTA”, an off-target search tool that implements many features, such as GC contents, RNA secondary structure, DNA methylation, epigenetic factors, etc., to predict the potential target sites. Although two and four potential target sites on chromosome 3 close to the large SV were found in β 2M-KO and APP knock-in clone, respectively, but these predicted sites were too far away from the large SVs to account for Cas9-cleavage. (**Supplemental Tables 3 and 7**).

Furthermore, we also used “Elevation” software to analyze the potential target sites. As shown in **Supplementary Table 8**, two potential target sites were predicted close to the large SV detected in the APP^{c/c} clone, but these predicted target sites are also too far (0.1- and 0.59 Mb, respectively) away from the large SV to account for Cas9 cleavage. However, when the Elevation software was interrogated for β 2M gene (ENSG00000166710), the results showed “unable to retrieve data” (**Supplementary Table 4**). These results, which further support our conclusion that the large SVs are independent of the homologous targeting by sgRNA, have been incorporated and described (p. 5-6) in the revised manuscript as **Supplemental Tables 3, 4, 7, and 8**.

2° in silico predictions should be confirmed by at least one in vitro additional technique such as CIRCLE-seq, a highly sensitive off-target detection method, or alternatively GUIDE-seq which is a good choice for measuring off-target specificity ex vivo in a cellular context.

It was reported that the GUIDE-seq method requires the transfection of cells *ex vivo* with dsODN tags, which may lead to problems because the pluripotent stem cells that were used in our studies often display a robust DNA damage response and undergo apoptosis in response to high levels of free DNA ends¹.

To comply with the reviewer's request, we performed “CIRCLE-seq” to identify the genome-wide off-target cleavage sites. As shown in **Supplementary Fig. 1**, CIRCLE-seq read was not detected in chr3, where the two large SVs were detected in our studies. Therefore, the results of the CIRCLE-seq analysis are now included in the revised manuscript to support our conclusion (p. 11).

3° To further increase the evidence of a non-homologous off-target effect, the use of high fidelity cas9 would be optimal for the two targets. Indeed, the persistence of an off-target deletion after transfection of a high-fidelity cas9 would be very convincing to support the very interesting but atypical hypothesis of the authors.

Although many high-fidelity Cas9 variants have been successfully developed, they are far from perfect. For instance, the high-fidelity Cas9 variants that are generated usually exhibit remarkable fidelity, but their enhanced fidelity comes at the cost of severe reduction of on-target DNA cleavage²⁻¹¹. Thus, the ideal Cas9 variants with both high fidelity and efficiency are still being sought-after. Therefore, it is very difficult to choose an ideal high-fidelity Cas9 to comply with the request.

Minor

If the three proposed experiments confirm the non-homologous off-target mechanism, it would be important in the discussion to propose mechanistic hypotheses.

As mentioned in the Discussion of the revised manuscript, the comprehensive genomic analysis reveals the existence of the unexpected, non-homologous off-target large structural variants after CRISPR-Cas9 mediated genome editing. It is intriguing that both of the two large SVs identified in these independently derived CRISPR-edited iPSC clones are located in *chromosome 3p*, which is distinctly apart from chromosome 15 (*β2M*) and 21 (*APP*), where the target genes reside. While Chromosome 3 spans about 198 Mb, the two large SV identified are ~1.56 Mb apart. Whether the relative proximity of these large SV on two unrelated CRISPR-edited iPSC clones is purely co-incidental or related to some unknown mechanisms that predispose this region of chromosome 3 to genetic alterations awaits further studies. To comply with the request, we added some discussions in this respect in the revised manuscript (p. 13).

Reviewer #2 (Remarks to the Author):

Tsai et al in “Comprehensive genomic analysis reveals atypical non-homologous off-target large structural variants induced by CRISPR-Cas9-mediated genome editing” describe their work using whole genome sequencing and optical mapping to determine if CRISPR/Cas9 editing creates structural variants in iPSC lines. The data and conclusions are clearly presented but the findings are expected based on the prior literature and thus the impact of the work is lessened. It is already well known that one must determine the genomic integrity of both engineered and un-engineered iPSC lines over time because of the risk of structural variants. This focused study confirms that need.

Specifically, I have the following concerns:

1. The term “comprehensive” overstates their work. They did not do RNA expression analysis nor chromatin analysis, as just two examples that the analysis was not comprehensive. They simply did whole genome sequencing.

As suggested by the reviewer, the title has now been changed to “Whole genomic analysis reveals atypical off-target large structural variants induced by CRISPR-Cas9-mediated genome editing”.

2. Would the variants have been identified by using simpler and cheaper SNP microarrays or array CGH?

In response to the reviewer’s comment, we performed the CytoScan high-density (HD) SNP array to provide the broadest coverage and highest performance for detecting chromosomal aberrations. CytoScan HD has greater than 99% sensitivity and can reliably detect 25-50 Kb copy number changes across the genome at high specificity with single-nucleotide polymorphism (SNP) allelic corroboration. As shown in the figure below, the three heterozygous large DNA deletions reported in our studies can be detected by the CytoScan HD array. The array CGH or SNP array is powerful in identifying losses and gains of genetic material. However, it cannot detect “balanced” SVs, such as translocation, inversion, and isochromosome formation. Therefore, SNP array and array CGH cannot fully address the issue of structure variation analysis.

Affymetrix CytoScan HD array analysis, including weighted log₂ ratio and smooth signal. (A) Detection of copy number lost at chr3:41,539,190-41,676,570 in the B2Mko-2 clone. (B) Detection of copy number lost at chr15:44,709,562-44,775,777 in the B2Mko-2 clone. (C) Detection of copy number lost at chr3:39,865,884-39,972,873 in the 71-APP^{C/C} clone. Open arrowheads denote the region of copy number lost.

3. The use of plasmid DNA to deliver the CRISPR-Cas9 is not state of the art. It is well known that the probability of generating off-target changes is increased with prolonged nuclease expression. The authors should use RNP delivery and see if the same thing happens.

To address the issue whether *RNP delivery* generated off-target changes as plasmid DNA, we consulted and discussed in person with Professor Akitsu Hotta from CiRA Kyoto University, who had previously used RNP delivery to perform the targeted disruption of HLA-A and -B genes¹². He performed the genome editing simultaneously of HLA-A and -B genes located on Chr 6 with the RNP delivery system. In his original report, chromosome karyotyping revealed that one of the 15 selected iPSC clones had translocation of chr 2 distal to 2p16-22, to the terminal end of the long arm of chr 15. With his consent, we employed the Cas-OFFinder and CRISTA algorithms to examine the potential off-target sites close to the chromosome region with mismatch numbers equal to or less than six, and the DNA bulge size equal to or less than two. As it turned out, we did not detect any off-target site close to this translocation region (**Supplementary Table 10 and 11**). These results suggest that this large chromosome translocation detected by karyotyping is also an unexpected, atypical non-homologous off-target large SV induced by CRISPR-Cas9 using RNP delivery, although the fact that this off-target large SV site without sequence similarity to the sgRNA, occurred during RNP genome editing was not mentioned in the original paper¹². In addition, the finding of the translocation of this big chromosomal fragment was detected by chromosome karyotyping, which has the limitation for detecting rearrangements with less than 5 Mb of DNA, although the use of multicolor fluorescence *in situ* hybridization can improve the resolution to about 100 Kb to 1 Mb in size. To detect the possible presence of other atypical large SV too small to be discerned by these techniques, we had originally planned to further examine the genomic integrity of Dr. Hotta's clones derived by RNP delivery of CRISPR-Cas9 by optical genomic mapping. Unfortunately, according to Dr. Hotta's reply, these specimens were no longer available. But, to address reviewers'

comments, we revised our manuscript to discuss the issue that non-homologous off-target large SV had also been found in clones induced by CRISPR-Cas9 using RNP delivery on chromosome karyotyping when SV is sufficiently large for detection (p. 14-15).

Reviewer #3 (Remarks to the Author):

Summary: Authors describe a methodology to detect on and off target genome changes using novel approach of combining optical genome mapping with linked read sequencing. this combination has been shown to be a comprehensive approach for detection of all classes of genome variants. The application of assessing genomic integrity in IPS cells is very important and pertinent today as cell therapies continue to make their way through drug development pipelines but without a consensus regarding how to ensure the absence of deleterious genome changes. In addition to describing the methodology, the authors also reported on unexpected large structural variants that presumably occurred due to CRISPR/CAS genome editing. These large SVs occurred in 2/11 samples bringing into question the safety of the current genome editing approach and underlining the critical need to standardize genomic QC in cell therapy applications.

Review overview: this manuscript presents the assessment of 11 edited cell lines and presents a case for the need to have better tools for genome assessment after editing. This is a topic of very high interest as there is significant debate regarding best practices and the need of higher resolution is gaining realization.

The study size is moderate but there very little published to date relating to SVs and edited cells and the findings that 18% of clones harbor a presumably editing induced large SV is quite striking and should have a high profile readership.

Some considerations that deserve attention are: linked read sequencing via 10X genomics is not commercially available any longer. Is there a viable substitute or alternative method?

Indeed, the linked read sequencing (10x genomics), the least expensive method to make high-accuracy one-step-from-chromosome-scale assembly, would not be available due to the legal dispute over patent rights. However, to our knowledge, other methods such as the UST's Transposon Enzyme Linked Long-read sequencing (TELL-SeqTM, UST¹³) that generates barcode linked-reads for genome-scale sequencing applications may be a viable option, although it is not clear how well this technique works. Alternatively, one of our authors, Dr. Pui-Yan Kwok, a worldwide expert in this area of research, is working on another alternative by combining high-accuracy short reads and PacBio-CLR (the continuous long read) to resolve structural variations and provide haplotype phasing. Other options include other long read sequencings such as PacBio and Nanopore sequencings; but these two long read

technologies have their own problems such as sequence accuracy and high cost. In our reply to the next comment (see below), we will have more discussions on this subject. These discussions will be incorporated in the revised manuscript (p. 13).

It would be good to discuss the complementarity of OGM and linked read sequencing for this application. In these results, both techniques found all reported variants so they provide value by orthogonal confirmation but what other complementarity is there? Could this work be done with just one technology or in a tiered fashion if there were time or price sensitivity, for example?

The linked-Read sequencing (up to 150 Kb) provides a clear advantage over short reads (typically 150-300 bp) alone, allowing for the construction of long-range haplotypes, and promising better long-range contiguity and resolution of repetitive regions. In contrast to DNA sequencing read, optical genome mapping (OGM) produces single long molecule (~225 Kb) maps which could cover heterozygous genome regions with different haplotypes that cannot be easily spanned by sequencing read. On the other hand, OGM detects SVs across the whole genome but does not provide sequence-level information or precise SV breakpoints. Therefore, combining linked-read sequencing and optical genome mapping would be a better option for genome integrity analysis. These considerations have been included in the "Discussion" section of the revised manuscript (p. 13).

In this report, we advocate the necessity to examine genome integrity after genome editing; therefore, both linked-read sequencing and optical genome mapping are used to support the conclusions. However, if there were time or price sensitivity issues, optical genome mapping may be the first option because this technology can detect SVs across the whole genome economically and with speed.

other points through the manuscript:

Intro - suggest to describe the standard methods for genome assessment (KT, CMA, NGS)

There are numerous methods used to analyze genomic alteration. Karyotyping is the oldest genetic method for chromosome alterations larger than 5 Mb; it can detect aneuploidy as well as transpositions, deletions, duplications, and inversions. Chromosomal microarray (CMA) is used to determine chromosomal imbalances such as amplifications and deletions called copy number variants (CNV). CMA provides submicroscopic resolution allowing us to visualize small regions that karyotyping cannot detect. Depending upon the particular array and how many DNA probes are used, it is possible to detect as small as 10 Kb. In contrast to microarray methods,

next-generation sequencing (NGS), also known as high throughput sequencing, directly determines the nucleic acid sequence of a given DNA. All of these will be included in the "Introduction" of the revised manuscript (p. 4-5).

line 86: this reference is not related to Bionano but is a predecessor tech, suggest to add nanochannel (bionano) and FGA references: Lam et al, Shieh et al?

Thanks. The correction was made.

line 133: "optical genome mapping, (OGM)" is more commonly used. suggest to change reference to Lam et al 2012

Thanks. The revision was made.

line 135: "due to much longer DNA molecule length used for analysis (up to 2.5 Mbp)"

As requested, the correction was made.

line 135: references are based on much older technology and 2 of them don't involve nanochannels. suggest: Lam et al, Mak et al, Chaisson et al or others related to SV detection by OGM using the Bionano platforms

Thanks. The revision was made.

Line 158 should be "a" and "b"

The correction was made.

line 214: specify whether "additional large SV at target site" is expected or not.

The revision was made (p. 13).

Figure 2 should be described better, these visuals are not well known and some panels, especially panel b and c might not be well understood.

The revisions were done as shown below and in **Fig. 2b-c** (p. 28).

Figure 2 (**b**) The large SV calls are constructed from linked-reads of the parental (NC01) and three single-cell clones of $\beta 2M$ knockouts. Peaks represent the predicted large SV calls compared to GRCh38. The asterisks * indicated large SVs on chromosomes 3 and 15 in $\beta 2M^{-/-}$, which were not detected in the NC01 and $\beta 2M^{-/-}$ 2 and 3.

In addition, (c) Matrix view of the overlapping barcodes analyzed with Loupe software (10x Genomics) showed heterozygous deletions in $\beta 2M^{-/-}$ on chr3:41437438–41837438 and chr15:44580927–44980927, respectively. The matrix view was plotted with the dark brown color representing the shared barcodes between two genomic segments marked on the X- and Y-axis. The X and Y axes correspond to the same genome region, so the barcode overlap matrix is symmetric. The diagonal shows the number of barcodes in each position along the displayed region. The colored band around the diagonal reflects long molecules that span several kilobases, thus generating barcode overlaps across their span. The color intensity drops suggest a relative drop in the number of molecules in that region. Therefore, the drop in coverage and the off-diagonal barcode overlap suggest a heterozygous deletion. The linear view represents the base coverage along the X-axis segment. The 136 and 68 Kb indicate heterozygous deletions. The figure legend has been revised accordingly (p. 28).

Supp figure 1: should be described better in the legend. Reader does not know what they are looking at or how they relate to SVs

The revisions were done as shown below and in **Supplementary Fig. 2.**

Supplementary Fig. 2. Large SV calls are constructed from linked-reads.

(a) The large SV calls are constructed from linked-reads of the parental (H9) and the four single-cell clones of $\beta 3galT5$ knockouts. (b) The large SV calls are constructed from linked-reads of the parental iPSC (ND40019*C) and the two single-cell clones of *LRRK2* (*G2019S*) knock-in. Peaks represent the predicted large SV calls compared to GRCh38. There is no difference detected between parental and mutants.

References:

- 1 Malinin, N. L. *et al.* Defining genome-wide CRISPR-Cas genome-editing nuclease activity with GUIDE-seq. *Nat Protoc* **16**, 5592-5615, doi:10.1038/s41596-021-00626-x (2021).
- 2 Bravo, J. P. K. *et al.* Publisher Correction: Structural basis for mismatch surveillance by CRISPR-Cas9. *Nature* **604**, E10, doi:10.1038/s41586-022-04655-8 (2022).
- 3 Casini, A. *et al.* A highly specific SpCas9 variant is identified by in vivo screening in yeast. *Nat Biotechnol* **36**, 265-271, doi:10.1038/nbt.4066 (2018).
- 4 Chen, J. S. *et al.* Enhanced proofreading governs CRISPR-Cas9 targeting accuracy. *Nature* **550**, 407-410, doi:10.1038/nature24268 (2017).

- 5 Huang, X., Yang, D., Zhang, J., Xu, J. & Chen, Y. E. Recent Advances in Improving Gene-Editing Specificity through CRISPR-Cas9 Nuclease Engineering. *Cells* **11**, doi:10.3390/cells11142186 (2022).
- 6 Kim, N. *et al.* Prediction of the sequence-specific cleavage activity of Cas9 variants. *Nat Biotechnol* **38**, 1328-1336, doi:10.1038/s41587-020-0537-9 (2020).
- 7 Kulcsar, P. I. *et al.* Crossing enhanced and high fidelity SpCas9 nucleases to optimize specificity and cleavage. *Genome Biol* **18**, 190, doi:10.1186/s13059-017-1318-8 (2017).
- 8 Kulcsar, P. I., Talas, A., Ligeti, Z., Krausz, S. L. & Welker, E. SuperFi-Cas9 exhibits remarkable fidelity but severely reduced activity yet works effectively with ABE8e. *Nat Commun* **13**, 6858, doi:10.1038/s41467-022-34527-8 (2022).
- 9 Kulcsar, P. I. *et al.* Blackjack mutations improve the on-target activities of increased fidelity variants of SpCas9 with 5'G-extended sgRNAs. *Nat Commun* **11**, 1223, doi:10.1038/s41467-020-15021-5 (2020).
- 10 Schmid-Burgk, J. L. *et al.* Highly Parallel Profiling of Cas9 Variant Specificity. *Mol Cell* **78**, 794-800 e798, doi:10.1016/j.molcel.2020.02.023 (2020).
- 11 Vakulskas, C. A. *et al.* A high-fidelity Cas9 mutant delivered as a ribonucleoprotein complex enables efficient gene editing in human hematopoietic stem and progenitor cells. *Nat Med* **24**, 1216-1224, doi:10.1038/s41591-018-0137-0 (2018).
- 12 Xu, H. *et al.* Targeted Disruption of HLA Genes via CRISPR-Cas9 Generates iPSCs with Enhanced Immune Compatibility. *Cell Stem Cell* **24**, 566-578 e567, doi:10.1016/j.stem.2019.02.005 (2019).
- 13 Chen, Z. *et al.* Ultralow-input single-tube linked-read library method enables short-read second-generation sequencing systems to routinely generate highly accurate and economical long-range sequencing information. *Genome Res* **30**, 898-909, doi:10.1101/gr.260380.119 (2020).

Reviewers' Comments:

Reviewer #1:

Remarks to the Author:

The manuscript by Liu et al is greatly enhanced by the experiments performed during revision. Several techniques (CRISTA and Elevation software and Circle seq) strengthen the evidence of this non homologous dependent mechanism of deletion. The authors have answered most of the points concerning the extra off-target analysis. The work is now much more convincing. Overall, I think the manuscript is an excellent contribution and warrants publication. The answers suit me and I have no other request for me.

Reviewer #2:

Remarks to the Author:

Tsai et al in the revision of "Whole genomic analysis reveals atypical non-homologous off-target large structural variants induced by CRISPR-Cas9-mediated genome editing" do a little work in addressing the concerns of the reviewers but there remains a major fatal flaw in the work. They failed to perform an identical analysis on the same number clones derived from the original iPSC line that were cultured and handled identically that were NOT exposed to Cas9 nuclease. That is, the authors have not shown that the SV's they found were not a spontaneous occurrence in this cell line when subjected to cloning and prolonged expansion and proliferation. Thus, the major novel conclusion that Cas9 can induce SVs at sites without sequence homology is not currently supported by the experimental data presented.

The following are also remaining concerns:

1. In the response to a reviewer the authors claim that a high fidelity Cas9 with preserved on-target activity is not available. That is a factually incorrect statement as both Thermo-Fisher and IDT sell such a Cas9. The descriptions of these Cas9 variants are in the literature.
2. The authors change the title but retain the term "comprehensive" in the abstract, discussion and other places. The failure to comprehensively change the terminology shows a lack of attention to detail or lack of appreciation of the reviewer's concern.
3. They should annotate the SV's they find in their own work and in the work of others. Do the SV's affect genes that might be known to affect proliferation/cell death/cell signaling that might be predicted to give that clone a proliferative advantage?
4. The authors were asked indirectly whether the site of the found structural variant was a fragile site. They did not seem to pick up on the question.
5. It seems the same SV does not occur in another iPSC cell line when exposed to Cas9 RNP. The authors need to repeat the experiment in another iPSC line using their technique with clones exposed and not exposed to Cas9 to determine if their findings might not simply be an artifact of the cell line they are using. That is, they need to test generalizability of the finding.

Reviewer #3:

Remarks to the Author:

The authors have sufficiently addressed the concerns that I have pointed out.

This work will make an important incremental contribution to the field for understanding the level of risk of unexpected structural rearrangements and technologies for detecting these changes.

Responses to the comments of the reviewers

Reviewer #1 (Remarks to the Author):

The manuscript by Liu et al is greatly enhanced by the experiments performed during revision. Several technics (CRISTA and Elevation software and Circle seq) strengthen the evidence of this non homologous dependent mechanism of deletion. The authors have answered most of the points concerning the extra off-target analysis. The work is now much more convincing. Overall, I think the manuscript is an excellent contribution and warrants publication. The answers suit me and I have no other request for me.

Thank you for the reviewer's positive comments that "the manuscript is an excellent contribution and warrants publication" and "The answers suit me, and I have no other request for me."

Reviewer #2 (Remarks to the Author):

Tsai et al in the revision of "Whole genomic analysis reveals atypical non-homologous off-target large structural variants induced by CRISPR-Cas9-mediated genome editing" do a little work in addressing the concerns of the reviewers but there remains a major fatal flaw in the work. They failed to perform an identical analysis on the same number clones derived from the original iPSC line that were cultured and handled identically that were NOT exposed to Cas9 nuclease. That is, the authors have not shown that the SV's they found were not a spontaneous occurrence in this cell line when subjected to cloning and prolonged expansion and proliferation. Thus, the major novel conclusion that Cas9 can induce SVs at sites without sequence homology is not currently supported by the experimental data presented.

Thank you for raising both old and new important issues. Please read our replies and findings of the new experiments described here and in the revised manuscript.

The following are also remaining concerns:

1. In the response to a reviewer the authors claim that a high fidelity Cas9 with preserved on-target activity is not available. That is a factually incorrect statement as both Thermo-Fisher and IDT sell such a Cas9. The descriptions of these Cas9 variants are in the literature.

This comment was originally raised by reviewer #1. We sincerely apologize for the confusion caused by our responses in our first revision. Here we would like to clarify that although high-fidelity Cas9 variants have been developed and are “available on the market”, they are far from perfect. Although high-fidelity Cas9 variants could exhibit remarkable fidelity, their enhanced fidelity comes at the cost of severely reducing the efficiency for on-target DNA cleavage [1-10]. Thus, the ideal Cas9 variants with both “high fidelity” and “efficiency” are still being sought-after. Therefore, it is not easy to choose a perfect high-fidelity Cas9 to comply with the request of reviewer #1. A study published in *Nature Biotechnology* suggests that the overall activity could be ranked as SpCas9 (wildtype) \geq Sniper-Cas9 > eSpCas9 (1.1) > SpCas9-HF1 > HypaCas9 \approx xCas9 \gg evoCas9, whereas their overall specificities could be ranked as evoCas9 \gg HypaCas9 \geq SpCas9-HF1 \approx eSpCas9 (1.1) > xCas9 > Sniper-Cas9 > SpCas9 (wildtype) [5]. Thus, it has also been suggested in several reports that for the development of high-fidelity Cas9 variants, the ideal and perfect Cas9 variants with both high “fidelity” and “efficiency” are still under active pursuits [7, 11].

2. The authors change the title but retain the term "comprehensive" in the abstract, discussion and other places. The failure to comprehensively change the terminology shows a lack of attention to detail or lack of appreciation of the reviewer's concern.

As suggested by the reviewer, we changed the term “comprehensive genomic...” to “whole genomic...” throughout the revised manuscript.

3. They should annotate the SV's they find in their own work and in the work of others. Do the SV's affect genes that might be known to affect proliferation/cell death/cell signaling that might be predicted to give that clone a proliferative advantage?

The large SV detected in the *B2M*^{-/-}-2 clone contains the *ULK4* gene; *ULK4* belongs to the family of unc-51-like serine/threonine kinases, which participates in a conserved pathway involving both endocytosis and axon growth [12-14]. Sequence variations in this gene have been associated with schizophrenia and bipolar disorder [15]. The large SV detected in the *APP*^{C/C} contains the *MYRIP* (Myosin VIIA and Rab interacting protein) gene, which is predicted to exhibit binding activity for actin and myosin, to be involved in positive regulation of insulin secretion and functions as a component of the exocytosis machinery [16, 17]. Thus, neither *ULK4* nor *MYRIP* is

known to be directly involved in cell proliferation, cell death, and cell signaling, which might be predicted to give the proliferative advantage to the clones.

This is also true for *MEMO1* (mediator of ErbB2-driven cell motility), the gene in the translocation junction revealed in Prof. Hotta's study [18]. *MEMO1* regulates Her2-dependent cell migration and is involved in breast carcinogenesis via regulating insulin-like growth factor-I receptor-dependent signaling events [19, 20]. Therefore, in response to this new request, we revised the manuscript in the Discussion section (p. 14-15) to point out the lack of involvement of the proliferative advantage of the genes to account for the large SV reported in our studies.

4. The authors were asked indirectly whether the site of the found structural variant was a fragile site. They did not seem to pick up on the question.

The issue regarding whether the found SV was a fragile site was not raised explicitly in the initial review. However, to address this issue now, we browsed the chromosomal fragile sites on HumCFS, a database of human chromosomal fragile sites [21]. There are four known chromosomal fragile sites on the chr3: FRA3A (chr3:23900001-26400000), FRA3B (chr3:58600001-63700000), FRA3C (chr3:182700001-187900000), and FRA3D (chr3:148900001-160700000). On the other hand, the large SV in *B2M*^{-/-}2 is at the chr3:41535938-41680677, and the large SV in *APP*^{C/C} is at the chr3:39882164-39973392, neither of which are fragile sites. Therefore, the large SVs reported in this study are not in the known chromosomal fragile sites. To comply with the reviewer's request, we have added the above statement in the Discussion of the revised manuscript accordingly (p. 14). In addition, neither the translocation junction (Chr2p22) of Prof. Hotta's clone [18] belongs to the known fragile sites.

5. It seems the same SV does not occur in another iPSC cell line when exposed to Cas9 RNP. The authors need to repeat the experiment in another iPSC line using their technique with clones exposed and not exposed to Cas9 to determine if their findings might not simply be an artifact of the cell line they are using. That is, they need to test generalizability of the finding.

In response to the reviewer's comment, we performed the following optical genome mapping to compare various clones exposed and not exposed to Cas9 (see p.12-13 of revised manuscript). First, we used NC01 (the parental cell line of *B2M* knockout) and iPSC-71 (the parental cell of *APP*^{C/C} knock-in) reported in our studies to examine whether their single-cell clones isolated from the PiggyBac transposon-

mediated knock-in will display large SVs or not, after repeated cultures and handlings, without exposure to Cas9 nuclease. As shown in the revised manuscript, the average N50 of those molecules was 246 Kb (range: 225-275 Kb), the average mapping ratio mapped to reference genome from molecules was 81.5% (range: 78.7-83.5%), and the average effective coverage was 84.8x (range: 57.1-111.48x). When compared to the reference genome (GRCh38), although several large SVs were detected in parental (NC01) and *ETV2i2* knock-in genomes, none of these large SVs was found only in the *ETV2i2* gene knock-in but not in the parental genomes (NC01; **Supplementary Fig. 2a**). Similarly, we could not find any large SVs which are present only in the *NGN2* knock-in genomes but not in the parental iPSC-71 (**Supplementary Fig. 2b**) either. These results strongly indicate that no unexpected large SVs occurred in the genomes of PiggyBac transposon-mediated gene knock-in clones derived from these two iPSCs (NC01 and iPSC-71), even after repeated cultures and handling.

Moreover, we also examined the genome integrity of the Cas9-mediated *DSG2* (F531C) gene knock-in in iPSC-71 and found no unexpected large SV (**Supplementary Fig. 3a**). Therefore, these results suggest that gene editing of these two iPSC lines, which requires cloning and prolonged expansion and proliferation, does not always generate the large SVs as detected in the *B2M* knockout and *APP^{C/C}* knock-in clones; in other words, the unexpected large SVs detected were not simply artifacts of the cell lines used. These findings have been added to the Results section (p. 12-13) of the revised manuscript.

In addition, it should be pointed out that gene editing of the iPSC line with CRISPR-Cas9 does not always induce large SV at atypical non-homologous off-target sites in all gene-edited clones. For instance, the large SV was only observed in one of three *B2M* knockout clones. Thus, based on our findings, we do not intend to advocate the abandonment of this powerful genome editing tool. Instead, we propose a strategy for detecting and validating CRISPR-Cas9 genome-editing outcomes using linked-reads and optical genome mapping. (see p.17 of the Discussion in revised manuscript).

Reviewer #3 (Remarks to the Author):

The authors have sufficiently addressed the concerns that I have pointed out.

This work will make an important incremental contribution to the field for understanding the level of risk of unexpected structural rearrangements and technologies for detecting these changes.

Thank you for the positive comments.

References

1. Bravo, J.P.K., et al., Publisher Correction: Structural basis for mismatch surveillance by CRISPR-Cas9. *Nature*, 2022. 604(7904): p. E10.
2. Casini, A., et al., A highly specific SpCas9 variant is identified by in vivo screening in yeast. *Nat Biotechnol*, 2018. 36(3): p. 265-271.
3. Chen, J.S., et al., Enhanced proofreading governs CRISPR-Cas9 targeting accuracy. *Nature*, 2017. 550(7676): p. 407-410.
4. Huang, X., et al., Recent Advances in Improving Gene-Editing Specificity through CRISPR-Cas9 Nuclease Engineering. *Cells*, 2022. 11(14).
5. Kim, N., et al., Prediction of the sequence-specific cleavage activity of Cas9 variants. *Nat Biotechnol*, 2020. 38(11): p. 1328-1336.
6. Kulcsar, P.I., et al., Crossing enhanced and high fidelity SpCas9 nucleases to optimize specificity and cleavage. *Genome Biol*, 2017. 18(1): p. 190.
7. Kulcsar, P.I., et al., SuperFi-Cas9 exhibits remarkable fidelity but severely reduced activity yet works effectively with ABE8e. *Nat Commun*, 2022. 13(1): p. 6858.
8. Kulcsar, P.I., et al., Blackjack mutations improve the on-target activities of increased fidelity variants of SpCas9 with 5'G-extended sgRNAs. *Nat Commun*, 2020. 11(1): p. 1223.
9. Schmid-Burgk, J.L., et al., Highly Parallel Profiling of Cas9 Variant Specificity. *Mol Cell*, 2020. 78(4): p. 794-800 e8.
10. Vakulskas, C.A., et al., A high-fidelity Cas9 mutant delivered as a ribonucleoprotein complex enables efficient gene editing in human hematopoietic stem and progenitor cells. *Nat Med*, 2018. 24(8): p. 1216-1224.
11. Kim, Y.H., et al., Sniper2L is a high-fidelity Cas9 variant with high activity. *Nat Chem Biol*, 2023.
12. Pelkmans, L., et al., Genome-wide analysis of human kinases in clathrin- and caveolae/raft-mediated endocytosis. *Nature*, 2005. 436(7047): p. 78-86.
13. Tomoda, T., et al., Role of Unc51.1 and its binding partners in CNS axon outgrowth. *Genes Dev*, 2004. 18(5): p. 541-58.
14. Ogura, K., et al., *Caenorhabditis elegans* unc-51 gene required for axonal elongation encodes a novel serine/threonine kinase. *Genes Dev*, 1994. 8(20): p. 2389-400.
15. Lang, B., et al., Recurrent deletions of ULK4 in schizophrenia: a gene crucial for neurogenesis and neuronal motility. *J Cell Sci*, 2014. 127(Pt 3): p. 630-40.

16. Huet, S., et al., Myrip couples the capture of secretory granules by the actin-rich cell cortex and their attachment to the plasma membrane. *J Neurosci*, 2012. 32(7): p. 2564-77.
17. Waselle, L., et al., Involvement of the Rab27 binding protein Slac2c/MyRIP in insulin exocytosis. *Mol Biol Cell*, 2003. 14(10): p. 4103-13.
18. Xu, H., et al., Targeted Disruption of HLA Genes via CRISPR-Cas9 Generates iPSCs with Enhanced Immune Compatibility. *Cell Stem Cell*, 2019. 24(4): p. 566-578 e7.
19. Schotanus, M.D. and E. Van Otterloo, Finding MEMO-Emerging Evidence for MEMO1's Function in Development and Disease. *Genes (Basel)*, 2020. 11(11).
20. Sorokin, A.V. and J. Chen, MEMO1, a new IRS1-interacting protein, induces epithelial-mesenchymal transition in mammary epithelial cells. *Oncogene*, 2013. 32(26): p. 3130-8.
21. Kumar, R., et al., HumCFS: a database of fragile sites in human chromosomes. *BMC Genomics*, 2019. 19(Suppl 9): p. 985.